# Estimating QoE from Encrypted Video Conferencing Traffic

**DOI:** 10.3390/s25041009

**Published:** 2025-02-08

**Authors:** Michael Sidorov, Raz Birman, Ofer Hadar, Amit Dvir

**Affiliations:** 1School of Electrical and Computer Engineering, Ben Gurion University of the Negev, Be’er Sheba 8410501, Israel; birmanr@post.bgu.ac.il (R.B.); hadar@bgu.ac.il (O.H.); 2Department of Computer and Software Engineering, Ariel University, Ariel 40700, Israel; amitdv@g.ariel.ac.il

**Keywords:** deep learning, encrypted traffic, quality of experience, machine learning, video conferencing

## Abstract

Traffic encryption is vital for internet security but complicates analytical applications like video delivery optimization or quality of experience (QoE) estimation, which often rely on clear text data. While many models address the problem of QoE prediction in video streaming, the video conferencing (VC) domain remains underexplored despite rising demand for these applications. Existing models often provide low-resolution predictions, categorizing QoE into broad classes such as “high” or “low”, rather than providing precise, continuous predictions. Moreover, most models focus on clear-text rather than encrypted traffic. This paper addresses these challenges by analyzing a large dataset of Zoom sessions and training five classical machine learning (ML) models and two custom deep neural networks (DNNs) to predict three QoE indicators: frames per second (FPS), resolution (R), and the naturalness image quality evaluator (NIQE). The models achieve mean error rates of 8.27%, 7.56%, and 2.08% for FPS, R, and NIQE, respectively, using a 10-fold cross-validation technique. This approach advances QoE assessment for encrypted traffic in VC applications.

## 1. Introduction

As we witness a significant increase in video content consumption on the internet over the past two decades, the ability to analyze this traffic, and in particular the QoE of the end user, is becoming critical. Internet service providers (ISPs) are placed in a unique position where they can influence the QoE of the end user directly for example by temporarily increasing the bandwidth (BW) of “content-hungry” network segments, prioritizing multimedia content over other traffic to improve experience through the use of a delay-sensitive application or by caching data in a content delivery network (CDN), bringing the content closer to the users.

To gather statistical information about the video being consumed by end users and assess its quality, ISPs traditionally relies on deep packet inspection (DPI) mechanisms, which allow for analysis of the data transmitted over the channel. This procedure is only feasible if the packets are transmitted in plain text. However, with the wide spread of end-to-end encryption on the internet, opportunities to employ these techniques have diminished significantly. There are numerous works on overcoming this encryption and extracting useful features from data traffic despite encryption. For example, [1] used packet length to predict websites visited by users. Ref. [2] later extended this technique to webpage recognition. In this context, although DPI may still be used in certain cases by exploiting vulnerabilities in encryption algorithms, research in this domain is advancing rapidly, so the number of cases in which DPI may be used shrinks even more. For example, a recent study [3] has demonstrated a new encryption algorithm that facilitates image transmission as a highly unassociated collection of values, which make it very hard to interpret without the key used for decryption while still perfectly reconstructing the data at the end point. Consequently, in most cases, ISPs are unable to utilize DPI and must instead rely on alternative metrics measured on the channel, such as latency, packet size, timing of packet arrival, etc. While these metrics are somewhat related to the ultimate goal of assessing users’ QoE, establishing a direct connection between these parameters and subjective perception of the viewed content remains a complex task.

In recent years, many applications have migrated to mobile devices due to their increased availability and advancements in cellular connectivity, which now commonly achieves speeds of 25 Mbps downstream and 3 Mbps upstream [4]. Among these applications are video conferencing (VC) platforms such as Zoom. However, a significant drawback of using mobile devices for VC is that the camera often becomes the bottleneck. The small form factor of mobile devices necessitates smaller camera sensors, which can capture approximately ten times less light compared to the full-frame 36 × 24 mm sensors typically found in digital single lens reflex (DSLR) cameras [5].

The reduced light capture of smaller sensors results in several image quality issues, including lower resolution, reduced dynamic range, and increased visibility of blurring and noise. As image quality is a key determinant of a user’s QoE, substantial effort is required to address these shortcomings. Since these issues stem from physical limitations of the camera system, solutions must rely on computational approaches. Techniques such as super resolution (SR), multi-frame noise reduction (MFNR), and high dynamic range (HDR) compression are employed to enhance image quality by increasing resolution, reducing noise, and improving the dynamic range of images, respectively. While these methods enhance image quality, they also increase the file sizes of packages that have to be sent over the network by adding the information generated by these algorithm, exerting additional stress on the network. In addition, as shown in the recent study by [6], who used MobileNetv3 [7] to predict defects in tile images, mobile devices still lack the computational abilities to sustain high accuracy in comparison to models that run on stationary systems. Cell providers, by evaluating the QoE of their users, might influence the total available bandwidth on a regional basis to increase the QoE of users who participate in a VC meeting.

The image quality transmitted between participants in a VC session is further constrained by the bandwidth provided by ISPs. To ensure smooth communication without temporal resolution issues (e.g., stuttering)—another critical factor influencing QoE—the available bandwidth must accommodate the enhanced image sizes resulting from quality improvement algorithms. In addition, since the computation of QoE criteria may not always be carried out on the mobile device (in connection to the previously mentioned limitations of mobile devices)—it must be performed in the “cloud”, which not only increases the bandwidth demand, but also the timing, further affecting the real-time requirement; even though there have been significant improvements in the shortening of the computation time of cloud services, as shown by [8], it is not comparable with the delay that may be tolerated in real-time applications. This presents a challenge for ISPs, as evaluating image quality in encrypted traffic, as previously mentioned, is a complex task.

In this context, internet traffic prediction mechanisms have become increasingly relevant, prompting extensive research. For example, in [9], the authors compared the performance of long short-term memory (LSTM) [10] and gated recurrent unit (GRU) [11] models in predicting internet traffic generated by multiple-agent communication over the State University of Ceará’s network during a six-month period. Similarly, ref. [12] conducted a study aimed at forecasting network load resulting from the transfer of large databases. They employed a custom model based on an auto-encoder (AE) [13] combined with GRU, achieving impressive results on a real-world database.

Criteria that try to evaluate the QoE of some videos may generally be categorized into two classes, viz., subjective and objective. The main difference between them depends on the level of human participation in the process of scoring. While the criteria that correspond to the former category require full human involvement to acquire the QoE rating, criteria which fall into the latter category can function without it, which is useful for evaluating the performance of ML algorithms, especially in the course of the training procedure of neural networks (NNs). On the other hand, since human perception does not always agree with mathematical models on which the objective criteria are based, subjective QoE measures present more realistic picture with regard to the satisfaction of the end user with the content. One of the most widely used subjective criteria of QoE is the mean opinion score (MOS), which was presented by the ITU-T Focus Group on IPTV [14] and represents the overall acceptability of an application (or a service) as it is perceived by the evaluating group of users. It is calculated by averaging the scores assigned by participants (usually on a scale of zero to five) to video samples.

With the notion of the limitations of objective QoE criteria, we are still inclined to use them in the assessment of QoE due to the ease of their application. Additionally, considering the sizes of the data required for the training of ML models, and especially NNs, mining subjective QoE metrics is infeasible.

Although objective criteria do not require human involvement in the process, they still have levels of complexity to them. This group of objective QoE criteria is divided into three subcategories, viz. full reference (FR), reduced reference (RR), and no reference (NR) metrics. While criteria that may be assigned to the first category use the original image and its full set of features as a reference to evaluate the corrupted image, criteria in the second category try to mitigate this dependence by using only the most important and easily acquired subset of features to predict the QoE. Lastly, methods that perform NR QoE evaluation are completely independent of the original image, using features from the channel for this matter. Consequently, we face a trade-off between precision and speed of application. While it is clear that evaluating original data is easier and may result in better accuracy, real-time applications such as VC require high responsiveness, making the acquisition of the reference image impractical in many cases.

Though many models for the QoE assessment of video streaming applications have been proposed over the past years, research in the field of VC has been fairly modest. Furthermore, since VC application QoE substantially differs from the QoE of streaming applications [15,16,17], there is a strong motivation for research in this direction. Finally, VC application use surged due to the COVID-19 pandemic, along with the “social distancing” phenomenon, [18,19], with an increase in adoption by many schools and organizations being recorded worldwide since then [20,21,22]; therefore, developing a QoE assessment model for VC applications is of high importance to this day.

## 2. Related Work

There are three main lines of previous work on predicting QoE in encrypted traffic, viz., methods based on statistical analysis, machine Learning (ML) algorithms (which mainly use decision trees or their variations), and deep learning (DL) algorithms employing artificial neural networks (ANNs), as summarized in Section 2. Here, we will discuss some of the most influential models in more detail (Table 1).

For estimating YouTube streaming quality over HTTPS, a model using the dynamic adaptive streaming over HTTP (DASH) protocol was proposed by Dubin et al. [31]. They showed that accurate classification of image quality is achievable just by looking into the data burst timings picked up from the channel. The authors achieved a high accuracy of 97% in classification tasks with these encrypted segments, placing them in one of the three categories based on the predicted resolution of the video being transferred in the segment.

Gutterman et al. [27] extended the work of Dubin et al. [31] by trying to predict two additional metrics for resolution—buffer warning and video state. In addition, in their study, resolution was predicted with higher granularity, extending the three categories in the work of Dubin et al. [31] to six. They trained a decision tree algorithm, achieving a scoring of 92.0% and 84.2% for the buffer warning and video state metrics; however, their performance in resolution predictions task was significantly lower than that of Dubin et al. [31], with just 66.9% accuracy. Even when the problem was reduced to a binary prediction (small and large resolution), the accuracy of their model did not exceed 91%.

Another model using decision trees was presented by Mazhar and Shafiq in [24]. In their experiments, they predicted three labels, viz., rebuffering events, video quality, and startup delay. Despite achieving a classification accuracy of 90% for HTTPS and 85% for QUIC traffic, their method has two significant drawbacks: low granularity (since they use binary class prediction alone for each parameter), and the inability to be used in online settings (as their model only predicts QoE retrospectively).

Orsolic et al. [23] proposed a framework for live QoE estimation named YouQ, which combines the proportions of the time spent in a high resolution and the duration of stalling events during video playback time. They evaluated their method against the MOS of human participants and tried to predict a binary label of a video segment (i.e., “high” or “low”), achieving an accuracy of 91%; prediction performance dropped to 84% after the addition of only a single extra class (viz., “medium”), thus highlighting the weakness in their approach.

Bronzino et al. [25] trained a random forest model to predict the startup delay and resolution of a video segment streamed with a QUIC protocol. Their dataset included sessions of four popular video streaming platforms, viz., YouTube, Amazon, Netflix, and Twitch. Though the proposed method predicted video resolution with an accuracy of 93%, it could not generalize well to other datasets.

One of the first studies investigating QoE in VC applications was performed by Chang et al. [18], in which the authors studied the influence of movement in the frame on the degradation in the quality of the image as a result of artifacts introduced by compression. They conducted 700 controlled VC sessions on three popular VC platforms (viz., Zoom, Webex, and Google Meet ), in which they calculated the point signal-to-noise ratio (PSNR), structural similarity index (SSIM), and pixel visual information fidelity (VIFp) of video fragments containing varying movement intensities as a measure of the QoE of these videos. The main limitation of this work, as pointed out by the authors themselves, was the lack of generality in their experiments; they used specific settings that may not always represent general use cases. In addition, they only performed their experiments on small-sized conferencing sessions (less than 11 participants), which may not be directly projectable onto medium and large conferencing sessions.

Yet another study was performed by Macmillan et al. [4], where the authors investigated differences in the minimum requirements for avoiding a decrease in QoE. They performed controlled experiments in which they manipulated the bandwidth of the down- and uplink channels, the number of participants in a VC session, and the viewing mode of the users (i.e., a single image of the speaker or a gallery) to compare the behavior of three popular VC applications (Zoom, Google Meet, and Teams) under changing network conditions. Interestingly, the authors showed that the viewing mode may reduce uplink utilization through the reduction in the resolution of the displayed image. One possible limitation of this work is its strong association with geographical location, as their entire dataset originated from the same University. This makes it unclear how their results will interpolate on sessions of users located in different geographical areas, where the round-trip time of the packets may not be neglected as it is in cases where users are all located relatively close to each other.

Various studies have employed models based on DL architectures. In this class of works, we recognize the study conducted by Shen et al. [29], in which the authors used a convolutional neural network (CNN) and round-trip time (RTT) in upstream traffic to predict the QoE of users on the YouTube and Bilibili platforms.

The paper in [32] offers a detailed analysis of how the proprietary Zoom application layer protocol works at the packet level. The authors identify unencrypted fields in the Zoom packet format as well as finding out how to group streams into meetings and how to identify peer-to-peer meetings. While the paper focuses on Zoom performance and behavior metrics under different network conditions, it does not infer end-user QoE. We do not focus on packet level analytics, but instead focus on predicting QoE metrics from the entire Zoom encrypted traffic stream using ML algorithms.

## 3. Methodology

For this purpose, based on previous work that was described in Section 2, we chose 6 classical machine learning models, and designed two custom DNN models. The latter were evaluated on a dataset of encrypted line pickup from controlled Zoom sessions, which were recorded in a lab environment. In this section, we will elaborate on each of the models used in this work, including the two custom DNN architectures.

### 3.1. Linear Regression (LR)

Linear regression is a statistical and machine learning method used for modeling the relationship between a dependent variable *y* and one (or more) independent variables $x_{i}$, assuming a linear relationship between them. Mathematically, a linear regression may be expressed as follows:(1)$$y=\beta_{0}+\beta_{1}x+\epsilon$$
where $\beta_{0}$ is the intercept of the line with the *x* axis, $\beta_{1}$ is the slope of the line, and the ϵ is the error term added to account for the noise in the data. In the multivariate case, Equation (1) turns into(2)$$y=\beta_{0}+\beta_{1}x_{1}+\cdots +\beta_{n}x_{n}+\epsilon$$
The goal of this model is to minimize the mean squared error (MSE) of the predicted and the true values, expressed as follows:(3)$$MSE=\frac{1}{n}\sum_{n}^{i=1}{\left(y_{i}-{\hat{y}}_{i}\right)}^{2}$$
making this model very simple to comprehend and analyze. However, this simplicity is also its main weakness, as few real-world models follow such a simple rule, though in many cases, it serves as a good baseline model. Another disadvantage of this model is its sensitivity to scale and outliers.

### 3.2. Decision Trees (DTs)

A decision tree is a supervised machine learning algorithm used for classification and regression tasks. It models decisions and their possible consequences as a tree-like structure, making it easy to interpret and visualize. The root node represents the entire dataset, while the intermediate nodes correspond to fractions of the dataset split based on a given decision rule. The leaf nodes (nodes without children) provide the output of the algorithm and represent the final prediction.

The algorithm works in phases, where in each phase the data are split based on a criterion (e.g., reduction of the variance, or the MSE, for regression problems, and Gini impurity or entropy for classification problems). The algorithm terminates when it reaches a predefined stopping criteria, such as maximum depth, with no significant information gain in the next phase, or if the number of nodes in the next phase will be smaller than the minimum number of desired nodes, etc. The prediction is performed by a majority vote for classification problems, and by mean or median calculation for regression problems.

Advantages of this method include interpretability, as it is very easy to visualize and understand the dependencies that arise from the DT. In addition, unlike LR, DTs may model complex, non-linear dependencies. Finally, this method is scale-agnostic, meaning that it does not require preprocessing of the data. On the other hand, DTs tend to easily overfit the training data, decreasing their generalization abilities. Furthermore, DTs may be unstable, and small changes in the training data may result in different tree structures. Finally, DTs are biased towards features with a multitude of distinct values, which may dominate the splits.

### 3.3. Random Forest (RF)

A random forest is an ensemble machine learning algorithm that combines multiple decision trees to improve accuracy and reduce overfitting. The training is carried out iteratively by training multiple DT models on separate subsets of the features, which are chosen randomly. It can be used for both classification and regression tasks, where the prediction in the first case is performed by majority voting on the DT classifiers, while in the latter case, it is achieved by averaging their outputs.

Advantages of this method include reduced overfitting (due to the randomness of its structure), high accuracy, its ability to model non-linear relationships, non-sensitivity to noise and outliers, and finally, its ability be used to rank features by their importance, which provides an additional layer of model interpretability.

Although accurate, this method may be too computationally intensive, especially when working with large datasets. In addition, due to aggregation of the RF, the inherent interpretability of the DT algorithm dissolves in the large number of small DT trainers, making the RF less interpretable in comparison with the plain DT algorithm. Finally, this method is sensitive to hyperparameter tuning and may still overfit if the hyperparameters are not chosen wisely.

### 3.4. Support Vector Machines (SVM)

Support vector machines, or SVMs, are a supervised machine learning algorithm used for classification and regression tasks. They are particularly effective in high-dimensional spaces and are known for their robustness in dealing with both linear and non-linear relationships.

The goal of an SVM is to find the optimal hyperplane that best separates the data into different classes. For linearly separable data, this hyperplane divides the data such that the margin between the classes is maximized, which is defined the distance between the hyperplane and the nearest data points from each class. SVMs iteratively seek to improve the margin and find such a hyperplane that the distance between it and each of the classes is maximized. The “support vectors” are the data points placed the closest to the hyperplane, and they determine its position and orientation. Mathematically, SVM algorithm may be expressed as follows:(4)$$\frac{1}{2}{\left||w|\right|}^{2}s.t.,y_{i}\left(w\cdot x_{i}+b\right)\geq 1$$
where *w* is the weight vector, *b* is the bias, and $y_{i}$ is the class label (±1).

The main advantages of this method are its high performance handling high-dimensional data, robustness to overfitting, and versatility in being able to handle both linear and non-linear datasets. On the other hand, similarly to RF, SVMs are also very sensitive to hyperparameter tuning, are computationally intensive, and lack interpretability.

### 3.5. Adaptive Boosting (AdaBoost)

Adaptive boosting (or AdaBoost for short) is another ensemble learning technique designed to improve the performance of weak learners, typically DTs. However, unlike the RF algorithm, which simply aggregates predictions and improves simple DTs through randomization, AdaBoost assigns weights to each DT “weak learner”, decreasing them on each iteration for learners closer to the label and increasing them for learners that score further away from it, making them more significant for the next weak learner. The final prediction is calculated based on majority voting of the weak learners for classification, or the mean of their predictions for regression problems.

The benefit of this method is its accuracy. In addition, many weak learner algorithms can be used with AdaBoost, making it very versatile. In addition, feature importance scores may be assigned to obtain insights into the data.

Conversely, this method is sensitive to noisy data and outliers due to the problem of overemphasizing misclassified samples, which may lead to overfitting. Since AdaBoost is a sequential model, it is inherently prone to consuming more time to converge compared to models that operate in parallel. Finally, the problem of interpretability common to all ensemble learning techniques is valid for AdaBoost as well.

### 3.6. Extreme Gradient Boosting (XGBoost)

XGBoost is an optimized and scalable gradient-boosting algorithm for supervised machine learning tasks, such as regression, classification, and ranking. It is widely used for its speed, accuracy, and versatility in handling structured/tabulated data. XGBoost builds an ensemble of decision trees and improves their performance iteratively by minimizing loss functions (e.g., MSE) via the gradient decentness algorithm. To reduce overfitting, XGBoost employs an $L_{1}$ (lasso) and $L_{2}$ (ridge) regularization, making it more robust than traditional boosting algorithms. In addition, it prunes trees that are too deep in order to keep the algorithm more compact and increase generalization. Another feature of XGBoost is its automatic handling of missing data in the process of training. Finally, unlike AdaBoost, XGBoost hyperparameters can be tuned in the process of training via a built-in cross-validation functionality.

The operation of XGBoost resembles that of AdaBoost, as it too starts by performing a random weight assignment and builds DTs, followed by iteratively correcting errors from the previous weak workers and assigning higher weights to poorly predicted samples. Finally, for the final prediction, the algorithm outputs an aggregation (sum for regression, and majority vote for classification) of all the weak learners.

This method is highly accurate and scalable (able to handle large datasets). In addition, it is flexible enough to support various loss functions, and uses built-in regularization functions to reduce overfitting.

The main disadvantage of this algorithm is its computational cost, which can rise proportionally to the size of the dataset. It suffers from interpretation problems, just like all the ensemble methods, and requires hyperparameter engineering like AdaBoost, despite its built-in optimization mechanism.

### 3.7. QoENet1D

The QoENet1D model is built from *N* chained blocks, where each block includes three elements, viz., a fully connected (FC) layer (with *U* units), a batch normalization (BN) [33] layer, and an activation (A) function, as presented in Figure 1. For all the intermediate layers, we chose the SiLU [34] activation function, which is known for its fast convergence properties; we normalized the features prior to the training procedure to remove the bias induced by their scale (i.e., transformed them to have a mean of 0 and a standard deviation of 1). Consequently, in the last prediction layer, we had to use an activation that was able to output values in the range of [−1,1], so the final layer in the model uses a hyperbolic tangent (Tanh) activation function, which fits this requirement. Between the blocks, we used skip connections, as this has been shown to improve the performance of deep neural networks [35].

### 3.8. QoEAE1D

The second model we propose comprises an autoencoder NN, which is responsible for projecting the features to a more informative latent space, and a feed-forward neural network used for the generating the final prediction. A schematic representation of the network is presented in Figure 2. Both the encoder (E) and the decoder (D) include the same number of layers and have the same architecture as that described in Section 3.7. The number of layers in this model may be set arbitrarily and constitutes one of the hyperparameters of the model; each layer in the downstream (encoder) reduces the number of neurons by 2, while the layers of the upstream (decoder) increase them by the same factor until the initial layer width is reached. Similarly to the QoENet1D Section 3.7, this model employs SiLU activation in all the layers of both the encoder and the decoder, while the final predictor block uses the Tanh activation function to fit the predictions into the region of [−1,1], which is necessary due to the standardization of the data.

## 4. Data

Conceptually, we utilized features that can be extracted from an encrypted channel in a VC session and have been designated by previous studies in the field as having a strong capacity for predicting QoE of end users. We evaluated the performance of our model on a custom dataset that imitated Zoom sessions under various channel bandwidths (changed with the NetLimitter application). It included network channel data pickup from 716 controlled Zoom sessions, where the features in Table 2 were collected and averaged over one-second time intervals to decrease the noise.

### 4.1. Data Collection

We collected a custom dataset that imitated Zoom sessions under various channel conditions. We chose the BW as the limiting parameter, as it is easy to control and interpret. In the data collection stage, we used various mobile devices to represent the clients, including smartphones and laptops of various brands and models, and a stationary computer located in our laboratory, which was designated for the role of the controller. The stationary computer was used to set the bandwidth to various values using the NetLimitter application (version 5.3.19.0) to simulate changing channel conditions, and to record the network pickup using the Wireshark application. A schematic representation of the experiment setup is presented in Figure 3, and an example of the extracted pickup file produced by the Wireshark application is shown in Figure 4. All the cameras in our experiments are listed in Table 3, and are standard cameras used in cellphones, laptops, and other non-professional photography equipment, lacking the full quality of image perception.

Using the Zoom Software Developer Kit (SDK), we extracted features (Figure 5) on the user side relating the data transmission in real time. The features are listed in Table 2 together with their data type and a short description of their functionality. All the features were averaged over one-second time intervals to decrease the noise.

### 4.2. Data Transformation

Due to the difference in the scale of the features extracted from the data stream, we normalized them around 0, with a standard deviation of 1, for use in the DNN learning. Following common practice, we removed the outliers that exceeded the $3\sigma_{x}$, which resulted in loss of only 15 samples out of a total of 716 (constituting only 2.095% of the data). In Figure 6 and Figure 7, we present a histograms of the labels and the distribution of the normalized features, respectively.

### 4.3. Exploratory Data Analysis (EDA)

First, we performed a Pearson correlation [36] test on the extracted features, as presented in Table 4. This analysis highlighted the features that correlated with the target variables, viz., NIQE, R, and FPS. Pearson correlation is a measure of the influence that a feature *X* has on its counterpart *Y*, and is defined as follows:(5)$$\rho_{X,Y}=\frac{Cov[X,Y]}{\sigma (X)\sigma (Y)}$$

This measure is located within the range [−1, 1], where 1 points to a perfect positive correlation (i.e., an increase in one feature results in an increase in the other), while −1 signifies a total reverse correlation, accordingly. Features with a correlation closer to ±1 are the most interesting, as it indicates that the feature, or it reverse, has a strong effect on the predicted variable.

We set the correlation threshold at ±0.5 and picked only the features that exceeded this value, both in positive or negative correlation, with at least two out of the three target labels. From this analysis, we conclude that the most correlated features are BW, PPS, ATP, and PL. As for the first three features in, it is quite intuitive that an increase in their value will decrease the NIQE, as a higher bandwidth leads directly to better resolution and fps, which, in turn, improve the perceived quality of the video.

The ATP, which has a strong positive correlation with the NIQE, i.e., an increase in this feature leads to a worse QoE for the observer (as an increase in the NIQE value signifies a degradation in the QoE). This is also intuitively clear, as packages that arrive at a lower rate may indicate a lower FPS. However, the high negative correlation of the PL is less intuitive, but it has an explanation, viz., as we increase the bandwidth, the video compressor has the ability to place more data in each packet, increasing the FPS, and thus also increasing the QoE of the viewer.

Among the strongest predictive values for NIQE, R, and FPS, we see BW, PPS, ATP, and PL; however, these features also exhibit strong inter-correlation between them, as shown in Table 4. To test their predictive abilities, we trained the models on different subsets of the features, and summarized the results in Table A2 and Table A3.

Another popular method for data analysis and the evaluation of information present in datasets is principal component analysis (PCA) [37]. It projects the data into a different subspace, where the regular axes (e.g., x, y for 2D, or x, y, and z for 3D, etc.) between which the data are located are linearly transformed into another set of axes—the principal components (PCs)—where the features express the highest variance in their distribution. The technique is performed iteratively, and the PCs are ordered from the one with the most variance to the one with the least. Finding the first one constrains the rest of the PCs, as they must all be uncorrelated with each other. The PCA algorithm implemented is an expectation maximization (EM) algorithm, where an initial PC (usually random) is produced at the beginning of the algorithm. Then, it proceeds in two alternating steps: calculation of the mean distance of all the data points from the proposed PC, and calculation of the gradient step in the direction of the minimization of this distance. In Figure 8, we present the variance that is added with the inclusion of each new PC. As presented in Table A1, we may conclude that it will be enough to use as few as 5 PCs to retain over 96% of the information in the data, while adding two more PCs (seven instead of the nine PCs) will increase the information to over 99%. From this analysis, we conclude that by transforming the features with the corresponding loadings, we can use as few as five features instead of the nine original features and still retain 99% of the information present in the data.

### 4.4. Feature Selection

As may be noticed in Table 4, some of the features exhibit high correlation (positive or negative) among themselves, e.g., BW and ATP (-0.93), PPS and PL (0.72), etc. To test the importance of the inclusion of highly correlated features together and the predictive advantage that each of these features induce on the final result, we conducted controlled experiments (validated with a 10-fold cross-validation technique), where we trained the models on subsets of the features with correlations exceeding ±0.5 and measured their performance. The final results of this analysis are summarized in Table A2 and Table A3.

From our analysis, we conclude that the most influential feature for many algorithms is latency (L), though QoENet1D and QoEAE1D models require all the features to converge. In addition, we may also conclude that there is no single set of features that results in a minimal error for all the models. This is to be expected, as different models exhibit certain vulnerabilities that may or may not be exhibited by the different datasets or different combinations of their features.

### 4.5. Feature Importance

To further validate the results achieved in Section 4.4, we performed two tests for feature importance, viz., coefficient and permutation tests. The essence of the first test is in the investigation of the weights associated with each of the features after fitting the model to the full dataset. In the second test, each of the features is shuffled (or permuted) within its column, and the degradation in the performance of the model serves as an indicator of the importance of this feature, i.e., the most important feature is the one that results in the higher degradation of performance. Each of the tests are performed with a 10-fold cross-validation method to ensure its statistical soundness.

From the analysis results of the coefficient test, which are presented in Figure A1, Figure A3, and Figure A5, we see that the findings support the results from the correlation test we conducted in Section 4.3, and the four most influential features are indeed the ones we found to have the highest correlation with the labels, viz., BW, PPS, ATP, and PL.

This was not the case with the results of the permutation test, as shown in Figure A2, Figure A4, and Figure A6). Although most of the BW, ATP, PPS, and PL still take one of the top four places in the importance ranking in all the cases, their order is different from the order that appears in the coefficient test; moreover, some of them even appear as the least influential (e.g., PPS in the AdaBoost model for R prediction, as shown in Figure A6). This phenomenon demonstrates that in our case, there may not be a single feature set that will fit all the models, and that the feature set must be chosen separately for each of them based on their performance evaluation.

## 5. Video QoE Assessment Methods

Methods for QoE estimation of video use quantitative values which are termed video quality assessment (VQA) scores. There are two main categories for video quality assessment, viz., subjective and objective. Methods which fall into the former category require the direct involvement of human participants, while methods from the latter category try to model human assessment, producing near-objective VQA scores in an automated manner.

On the other hand, objective methods for QoE assessment do not require human participants in order to produce a VQA score, though they require a reliable model that sufficiently suits the QoE function of a given customer group. In [38], the authors compare five objective QoE parameters: the point signal-to-noise ratio (PSNR), video quality metric (VQM), structural similarity (SSIM) index, mean structural similarity (MSSIM) index, and visual signal-to-noise ratio (VSNR). They propose dividing them into three main categories: full reference (FR), i.e., metrics that require the whole original video in order to assess the quality of the distorted version; reduced reference (RR) QoE metrics, which require only a subset of features of the original video to produce the QoE assessment; and metrics that are completely independent of the original video, which are categorized as no reference (NR).

Just by looking at an image or a video, we can asses its quality fairly quickly. However, finding an objective criterion for quality assessment is a challenging task for an automated system. Many lines of work have tried to propose their criteria for a numeric quality measure, but they can all be divided into two categories, viz., full reference (FR) and no reference (NR) QoE metrics. The difference between the two may be summarized in the fact that while the former requires human assessment, the latter uses an algorithmic approach to achieve the quality criterion. In this work, we chose to concentrate on one metric from each category, i.e., the video multi-method assessment fusion (VMAF) [39] for the FR QoE metric, and the naturalness image quality evaluator (NIQE) for the NR metric [40].

### 5.1. Mean Squared Error (MSE)

The MSE of an image may be calculated as a pointwise calculation of the mean squared differences in pixel intensity of the original image *I* and a lossy image *L*, i.e.,(6)$$MSE\left[I,L\right]=\frac{1}{MN}\sum_{i=0}^{m-1}\sum_{j=0}^{n-1}{\left[I\left(i,j\right)-L\left(i,j\right)\right]}^{2}$$

### 5.2. Root Mean Squared Error (RMSE)

As the MSE squares the distances, we find a deviation in the case of measure units, which also become squared. To address this issue, and in case the measure units are present, we can obtain the square root from the MSE, which results in the RMSE, as shown in Equation (7)(7)$$RMSE\left[I,L\right]=\sqrt{\frac{1}{MN}\sum_{i=0}^{m-1}\sum_{j=0}^{n-1}{\left[I\left(i,j\right)-L\left(i,j\right)\right]}^{2}}$$

### 5.3. Point Signal-to-Noise Ratio (PSNR)

The PSNR metric is defined via the mean squared error (MSE) of the original image *I* and the lossy image *L*, as shown in Equations (11) and (12).(8)$$PSNR\left[I,L\right]=10log_{10}\left(\frac{MAX{\left[I\right]}^{2}}{MSE[I,L]}\right)$$

### 5.4. Structural Similarity (SSIM) Index

The authors in [41] base their criterion on the fact that the human visual system is more adapted to extracting structural features from a scene than comparing it pointwise, as is done in MSE or PSNR. Instead, humans form an opinion (e.g., enjoyment or annoyance) of an image based on certain features that are either present or absent.

The SSIM criterion is based on three aspects that, in the eyes of the authors of [41], best represent the HVS. They maintain that instead of pointwise assessment (as is done in MSE and PSNR), a distorted image should be compared to the original based on its *luminance*, *contrast*, and the *structure* of the items in it.

### 5.5. Mean Structural Similarity (MSSIM)

The authors in [41] state that the SSIM criterion does not perform well for large images due to spatial non-stationary property of natural images. Due to this limitation of the method, they propose applying it to small regions (i.e., 8×8 or 11×11 images), and calculating the statistics used in the SSIM criterion within each of them. Since it is more practically useful to have a single criterion for each image, they propose summarizing it as an arithmetic mean of the locally computed SSIM criterion, as shown in Equation (9)(9)$$MSSIM\left(X,Y\right)=\frac{1}{M}\sum_{j=1}^{M}SSIM\left(X_{j},Y_{j}\right)$$
where *X* and *Y* are the corresponding images being assessed, which are divided into *M* separate sub-images.

### 5.6. Visual Information Fidelity (VIF)

Visual information fidelity is an image quality assessment metric that measures the similarity between a reference image and a distorted image, making this method fall into the full reference method class. It quantifies the fidelity of the visual information retained in the distorted image by modeling how human vision perceives information. This criterion is grounded in the information theory by evaluating how much information from the reference image is preserved in the distorted image and perceivable by a human observer. It is accomplished through the incorporation of human visual system (HVS) models to assess quality based on how images are perceived by humans, rather than a purely mathematical representation such as MSE Section 5.1 or PSNR Section 5.3. VIF leverages the statistical properties of natural images to determine how well the distorted image retains these characteristics. For this purpose, images are analyzed in the wavelet domain, which allows separation into multiple frequency bands to better simulate human perception. Mathematically, VIF may be expressed as follows:(10)$$VIF=\frac{I_{d}}{I_{r}}$$
where $I_{d}$ is the information retained in the distorted image, while $I_{r}$ is the information of the reference image. VIF ranges from 0 to 1, where 0 signifies no information retention from the original image in the distorted image, while 1 signifies perfect information retention.

The advantage of this criterion is that it tries to mimic the HVS, which gives it an advantage in tasks where human-related features are assessed. In addition, it is robust against various types of noise, including blurring and compression artifacts. On the other hand, VIF is more computationally intensive compared to simpler algorithms, such as MSE Section 5.1 or PSNR Section 5.3. Finally, the dependence of this criterion on a reference image may prevent its use in cases where the reference image is unavailable, e.g., due to end-to-end encryption.

### 5.7. Video Multi-Method Assessment Fusion (VMAF)

This metric was developed by Netflix as an alternative to the peak signal-to-noise ratio (PSNR) metric which is defined via the mean squared error (MSE) of the original image *I* and the lossy image *L*, as shown in Equations (11) and (12).(11)$$MSE\left[I,L\right]=\frac{1}{mn}\sum_{i=0}^{m-1}\sum_{j=0}^{n-1}{\left[I\left(i,j\right)-L\left(i,j\right)\right]}^{2}$$(12)$$PSNR=10log_{10}\left(\frac{MAX_{I}^{2}}{MSE}\right)$$

The authors of [39] maintain that PSNR does not provide a reliable criterion for QoE assessment. They developed the VMAF as a machine learning (ML)-based metric, which trains a support vector machine (SVM) model on three basic metrics, i.e., two for image quality (visual information fidelity (VIF) [42] and the detail loss metric (DLM) [43]), and one for temporal quality assessment (motion), which performs a simple average of the absolute differences in pixel-wise intensities between two consecutive frames.

VMAF uses human assessments of video fragments corrupted with noise as labels along with the corresponding clean reference video fragments (i.e., without noise added). Human evaluators rate the corrupted video fragments on a scale of 1 to 5 (i.e., where 1 means “very annoying” while 5 is “not noticeable”). Then, the scores for each video from every evaluator are combined and normalized to form the differential mean opinion score (DMOS), which lies in the range of [0, 100], where a score of 100 given for the reference video. We extracted several FR video quality metrics from Zoom traffic. In order to assess the quality degradation caused by reduced bandwidths, we needed to compare a reference source video of original quality that was impacted by the reduced network conditions to the same video stream at the receiving end. With access to the video recorded by the Zoom application, we observed that when establishing a two-party conference call, Zoom reduces the quality of the recorded video at the source and at the destination at the same time. In order to obtain a good-quality reference, we managed to force the Zoom application to produce and record a good-quality video. This was accomplished by adding a third party to the call, with no resulting bandwidth limitation. In this way, the sending Zoom application allowed for recording a good-quality reference, which was not impacted by the restricted bandwidth of the second party added the call. The setup used to record the reference and the destination videos is depicted in Figure 9. In order to perform a reasonably accurate comparison between the two recorded streams, they have to be synchronized. We used the audio recording to synchronize the streams by creating an abrupt loud instantaneous noise (similar to “cut” used when recording a “video take” for a movie) and synchronized the videos at the sender and recipient ends using that noise. The results of the full reference video QoE metrics measured using the setup depicted in Figure 9 are provided in Figure 10. The slight improvement in quality observed when the bandwidth was decreased can be explained by the slight decrease in the Qp parameter performed by Zoom during the bandwidth drop transition period (described in Section 6.1.2). It is assumed that Zoom decreases the Qp to compensate for reduced quality during the bandwidth drop transition.

### 5.8. Naturalness Image Quality Evaluator (NIQE)

In our experiments, we used the NIQE metric proposed [40]. Due to the challenges of recording Zoom video sessions with FR and in order to make the dataset more accommodating to future expansions (which do not necessitate complicated setups and saving a reference for each recorded video clip), we researched the no reference (NR) QoE metric alternatives. A promising NR metric that has emerged is the naturalness image quality evaluator (NIQE) [40], which measures the distance of the frame from naturalness, with a smaller score indicating better perceptual quality. NIQE measures the distance between the natural scene statistics (NSS)-based features calculated from an image and the features obtained from an image database used to train the model. The features are modeled as multidimensional Gaussian distributions. We calculated NIQE as a function of bandwidth, resolution, and FPS. The corresponding graphs are depicted in Figure 11. As can be seen in Figure 11b,c, the NIQE is consistent with the available bandwidth and video frame resolution, respectively, such that the larger the bandwidth or resolution, the smaller the NIQE, thus indicating better quality. We can observe a range of resolutions for which the NIQE value remains constant, indicating that the image naturalness metric at these resolutions is sufficiently good despite the change. An interesting observation is depicted in Figure 11a, indicating the changes in NIQE with FPS, where the metric is measured per single frame and is therefore spatial by nature. The temporal resolution of the video frames has an impact on the spatial quality. This is due to the compression algorithm utilizing quantized residuals, which are determined by block prediction accuracy, which is, in turn, impacted by the magnitude of the motion vectors (MV) used for inter-prediction. The larger the movement between consecutive frames, the larger the MV magnitude and the less accurate the prediction, thus increasing the value of the temporary predicted block residuals and reducing the quality due to their quantization.

## 6. Experiments

### 6.1. Zoom Behavior in Response to Decrease in BW

From the captured dataset, described in Section 4.1 have inferred the behavior of the Zoom application as a function of different channel conditions, based on QoS features extracted from the line, as presented in Figure 12.

#### 6.1.1. Adaptation Cycle Due to Reduced Available Bandwidth

We shall refer to ‘transition’ as the period during which the Zoom application adjusts its video transmission parameters when a degradation is introduced into the channel quality, such as a drop in BW. We discovered that, during transition, Zoom goes through the following steps, which may take up to 10 seconds, as observed in our experiments: (1) *reducing video spatial resolution (lowering image quality)*; (2) *reducing video temporal resolution (lowering the FPS)*; and (3) *restoring spatial resolution to original value, but with a lower FPS, to compensate for the drop in the available BW*. This behavior makes sense, given that in typical video conferencing scenarios, there are only small spatial movements, meaning that users’ perceived quality will be impacted more by the spatial resolution than by the temporal resolution. The charts in Figure 13 illustrate the steady-state variation of FPS and L as a function of available BW (measured after waiting for the transition period to pass and the spatial resolution to be restored). Although these measurements were taken from a single session, they represent a consistent behavior that we have observed in multiple similar sessions. A dramatic drop from 25 fps to less than 10 fps is observed when the available bandwidth drops from 120 kbps to 60 kbps.

#### 6.1.2. Impact on Quantization Parameter (Qp)

Assuming that Qp may be used by Zoom as a dominant parameter to adapt to changing channel bandwidth conditions, and therefore may be used as a good indicator for QoE, we have explored the change in Qp when reducing the available BW. We observed that the Qp remains unchanged for I-frames and slightly decreases with reduced bandwidth for P-Frames, as shown in Figure 14. This indicates that the Zoom application tries to compensate for the reduced BW, which is accommodated by reduced FPS, by adjusting to a finer quantization of the P-Frames; however, the change is small, and we consider it too hard to estimate from encrypted traffic. Therefore, it is less valuable as a QoE predictor. The change in Qp with reduced bandwidth is illustrated in Figure 14.

### 6.2. Hyperparameter Selection

The hyperparameters of the QoENet1D and the QoEAE1D models were chosen by considering different subsets of layers L∈{8,16,32} and units U∈{8,16,32,64,128,256,512}, and initial learning rates were chosen from lr∈{0.01,0.008,0.005,0.001}. In addition, we checked the optimal number of training epochs E∈{10,50,100}. After extensive evaluation using a 10-fold cross-validation method, we arrived at the optimal architecture for both the QoENet1D and QoEAE1D; L=8,U=512,lr=0.01 was chosen as the architecture that achieved the best results. For both models, the number of training epochs was set to E=50; as in most of the experiments, the best loss was centered around E=39±10.

### 6.3. Training Procedure

All the classical ML models, viz., LR, DT, RF, SVM, AdaBoost, and XGBoost, were trained in a supervised fashion, without any manipulation of the data except for outlier exclusion, as described in Section 4.2. In models based on the NN architecture, data were standardized (i.e., $\frac{x-\mu_{x}}{\sigma_{x}}$) due to the scale sensitivity of the models. The training dataset was divided in a proportion of 80:20 for training and validation, respectively. In our experiments, we used a batch size of 128 samples, which were shuffled for the training data, but not for the validation or test datasets. The loss used in the training procedure was the mean squared error (MSE) loss, and we chose the Adam optimizer with standard settings as the optimizer.

## 7. Results

In this section, we will describe the results of the experiments we conducted to test the effectiveness of our methods. We used the dataset described in Section 4. In Table 5, we present the mean absolute error percentage of the predicted and actual value of the three labels in our dataset that are indicative of the QoE, viz., NIQE, R, and FPS, which is calculated as follows:(13)$$Err\left(t,p\right)=|100-\frac{p\cdot 100}{t}|$$
where *t* and *p* are the true and the predicted values, respectively. We see that the best-performing model is the RF. The two proposed models, viz., QoENet1D and QoEAE1D, while definitely exhibiting lower performances than that of the RF, exhibit significantly less variance than other models. Additionally, in the light of the size of the dataset (containing just 720 samples), it is impressive that the models succeeded in converging to results comparable with ML algorithms that may be fitted to a small dataset.

## 8. Conclusions and Future Work

We have explored the extraction of QoE metrics from encrypted Zoom video conferencing traffic. Using four classical ML algorithms—namely linear regression, decision trees, random forest, and SVM—along with two boosting algorithms, AdaBoost and XGBoost, which are specifically designed for tabular data and have emerged as predominant classification methods for VoD traffic, we adapted our approach to a regression problem instead of a low-resolution classification problemBy regressing the model to real-valued labels, we achieved a mean error of 2.08% for NIQE, 7.508% for FPS, and 7.607% for resolution prediction on our relatively small dataset of 720 samples, despite the high imbalance in the labels, as shown in Figure 6. Furthermore, we designed two custom deep neural network models, which we named QoENed1D and QoEAE1D, to fit the data. These models significantly improved variance in the errors while achieving competitive results compared to previous methods, despite the scarcity of data usually needed to train a DNN model. We have demonstrated excellent accuracy in predicting resolution, FPS, and the naturalness image quality evaluator (NIQE). We are presently working on enhanced automation that will allow us to capture larger datasets in order to drive a more robust and generalized prediction model. In addition, we plan to expand the scope of this research to other video conferencing tools, notably Microsoft Teams and Google GoToMeeting. We also plan to create a large dataset of video conferencing clips and perform a mean opinion score (MOS) experiment with human observers. We will then compare the results of the metrics to those of the MOS and perform classification for both. 

## Figures and Tables

**Figure 1 sensors-25-01009-f001:**
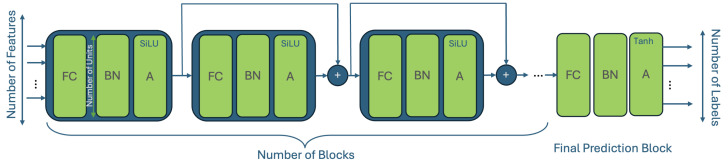
This figure presents a schematic representation of the proposed network, i.e., the QoENet1D; it is built from *N* blocks of consecutively stacked fully connected (FC) batch normalization (BN) layers and an activation function. The last prediction block uses the Tanh activation function, while all other layers have SiLU as an activation function.

**Figure 2 sensors-25-01009-f002:**
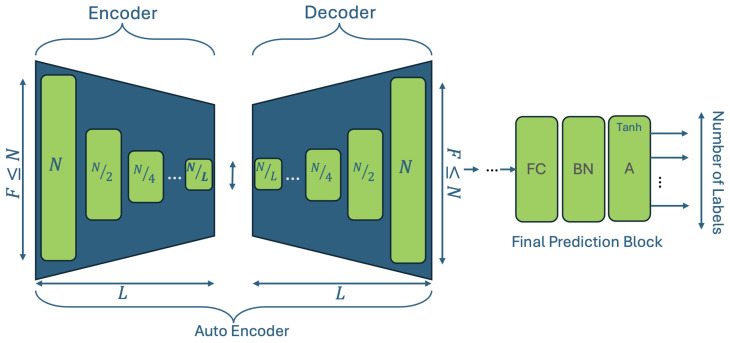
This figure presents a schematic representation of the QoEAE1D, which is based on an autoencoder network followed by a prediction block. The last prediction block uses the Tanh activation function while all other layers have SiLU as their activation function.

**Figure 3 sensors-25-01009-f003:**
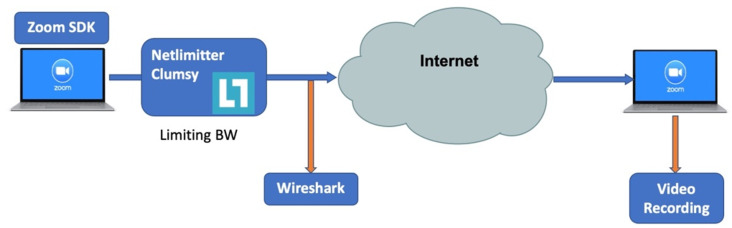
Setup used to capture Zoom traffic features and labels.

**Figure 4 sensors-25-01009-f004:**
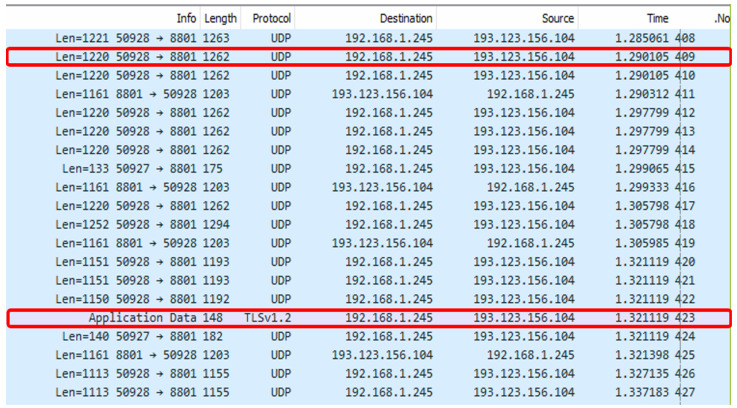
Example of the data collected with the *Wireshark*. In the top red box we see the UDP traffic corresponding to the conversation packets, while the bottom red box depicts the Transport Layer Security (TLS) packets being transmitted in the course of the conversation application.

**Figure 5 sensors-25-01009-f005:**
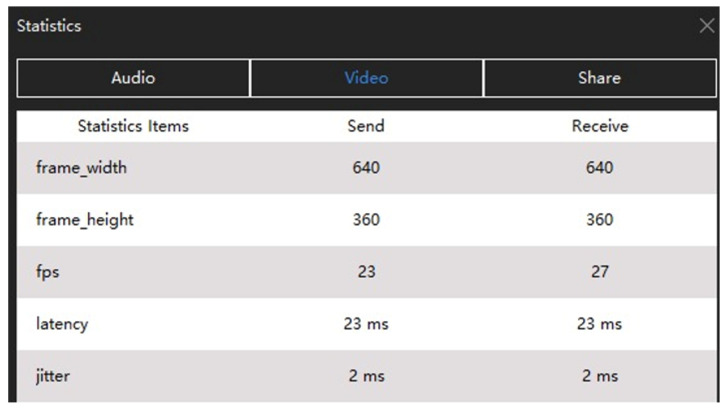
Example of the QoS features extracted from a Zoom API.

**Figure 6 sensors-25-01009-f006:**
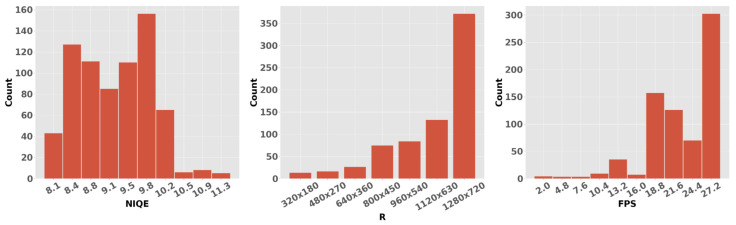
Histogram of the predicted labels, viz. NIQE, FPS an R.

**Figure 7 sensors-25-01009-f007:**
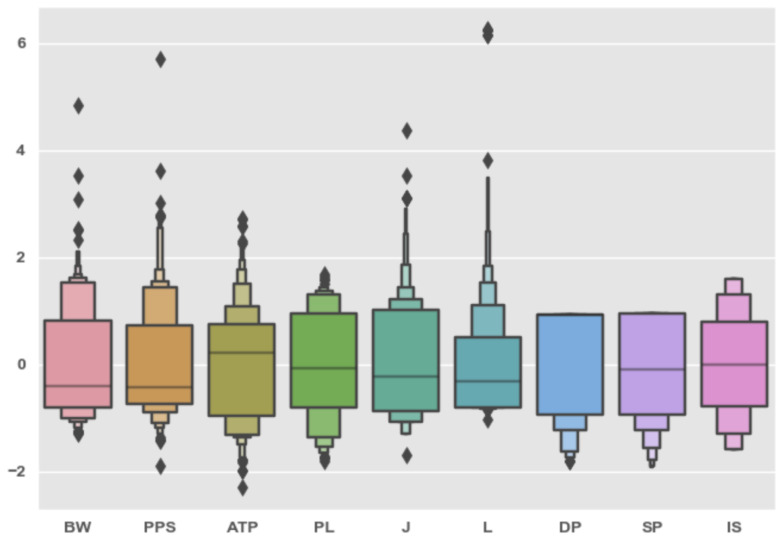
Distribution of the normalized (centered around 0 with standard deviation of 1) features extracted from the encrypted stream that we used to predict the NIQE, FPS, and the R labels.

**Figure 8 sensors-25-01009-f008:**
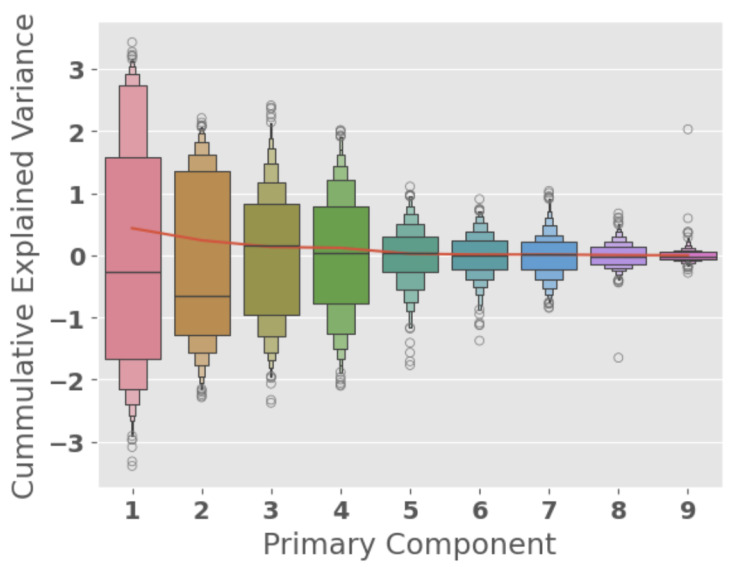
Principal component analysis (PCA) of the nine features in our dataset. Each PC is represented by a box plot, where the width of the box represents the quantity of values in each range. The circles on the top and bottom of the boxes represents outliers. The red line corresponds to the added variance by inclusion of the particular PC to the features.

**Figure 9 sensors-25-01009-f009:**
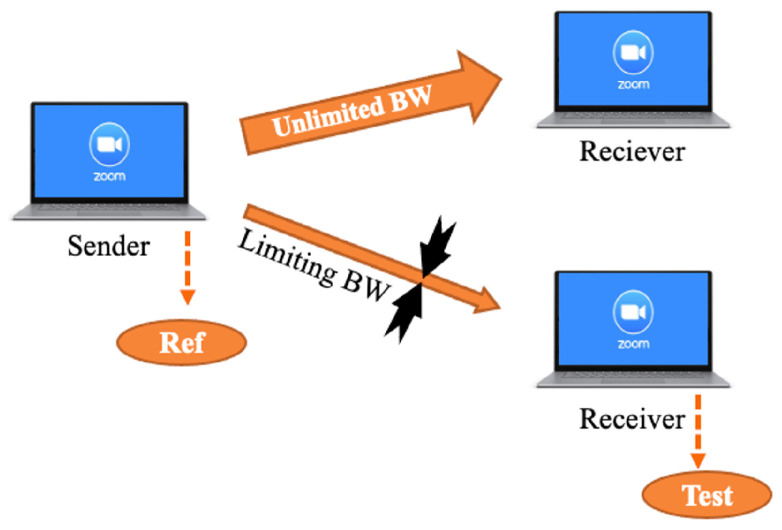
The setup for extraction of the VMAF metric.

**Figure 10 sensors-25-01009-f010:**
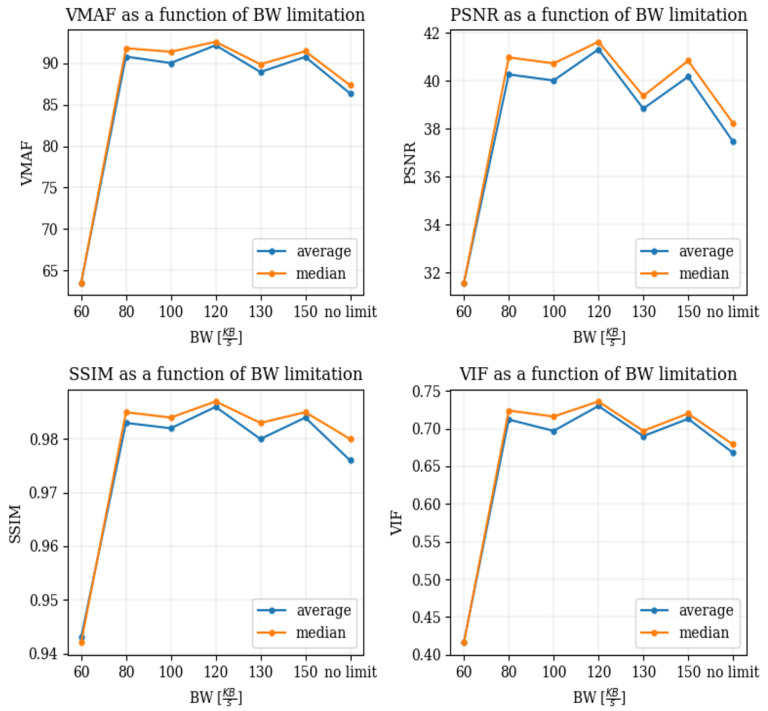
The change in the FR metrics as a function of the BW of the line.

**Figure 11 sensors-25-01009-f011:**
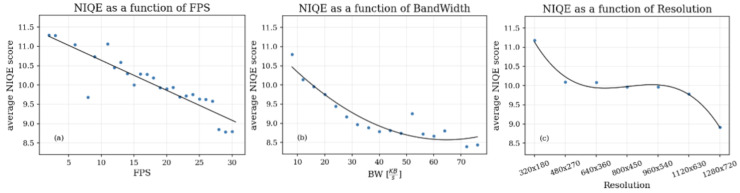
Behavior of the NIQE QoE metric as a function of FPS, BW, and R labels. As we can see from the figures, NIQE fits well the traditional QoE measures, which makes it a good fit as a candidate metric for QoE assessment.

**Figure 12 sensors-25-01009-f012:**
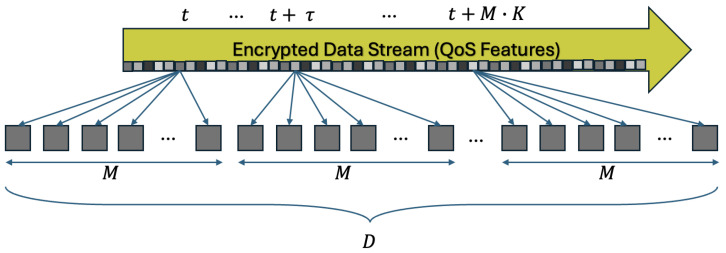
Schematic representation of data used for training the models.

**Figure 13 sensors-25-01009-f013:**
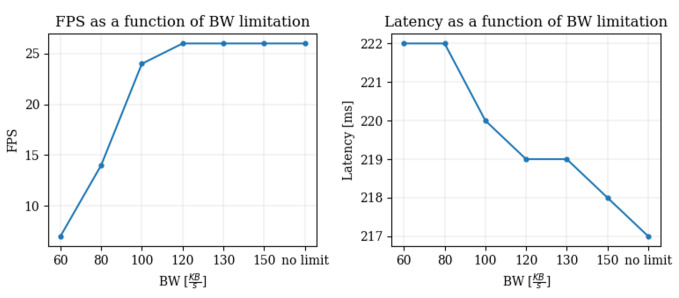
The adaptation pattern of the latency on the channel and the FPS of the video as a function of the BW.

**Figure 14 sensors-25-01009-f014:**
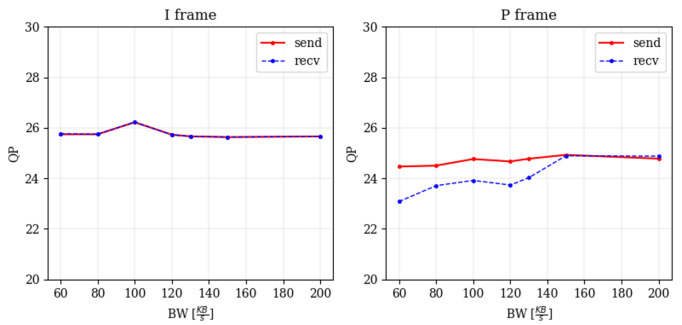
Qp variation as a function of the BW.

**Table 1 sensors-25-01009-t001:** Summary of the most influential works performed in the domain of QoE prediction in encrypted traffic.

Model	Reference
Decision Tree	Orsolic et al. [23]
	Mazar and Shafiq [24]
	Bronzino et al. [25]
Random Forest	Dimopolus et al. [26]
	Orsolic et al. [23]
	Bronzino et al. [25]
	Gutterman et al. [27]
Regression	Bronzino et al. [25]
Session Modeling	Mangla et al. [28]
DNN	Shen et al. [29]
	Oura et al. [30]

**Table 2 sensors-25-01009-t002:** The table below presents the names of the features extracted from the encrypted Zoom traffic, together with their corresponding data type, and description (data types with * signify categorical feature).

Name	Abbreviation	Data Type	Description
Bandwidth	BW	float32	The number of bits per second allowed to be transmitted on the channel
Natural Image Quality Evaluator	NIQE	int16 *	Objective, no reference QoE criterion
Resolution	R	int16 *	The number of pixels in the width and the length of the presented image
Frames per Second	FPS	int16	The number of frames per second in the playback
Latency	L	float32	Initial time in milliseconds after which the video starts
Jitter	J	float32	Variance of data packet arrival times measured in milliseconds
Packets per Second	PPS	int16	Number of packets arriving in each millisecond
Destination Port	DP	int16 *	The second layer port on the remote host where the transmitted data are directed
Source Port	SP	int16 *	The second layer port on the local host through which the data are transmitted
Average Time between Packets	ATP	float32	The time difference between the reception of two consecutive packets
Packet Length	PL	int16	Number of bytes in each transmitted packet
Interval Start	IS	int16	The time the video started after the initial

**Table 3 sensors-25-01009-t003:** Specifications of the sensors of the cameras used in the experiments.

Device	Resolution	Sensor Size (Inches)
Iphone SE (2020)	1.2 MP	1/3
Iphone 11	12 MP	1/2.55
Galaxy 9	8 MP	1/2.55
MacBook Air m1	720p	1/2.55

**Table 4 sensors-25-01009-t004:** Pearson correlation of different features in the dataset. In bold script are presented the features that have correlation ρ>0.5 with at leas two of the labels, viz., NIQE, R, and FPS, or with other features in the dataset.

	Features							Labels		
	L	J	PPS	DP	SP	ATP	PL	NIQE	R	FPS
**BW**	−0.27	−0.38	**0.95**	−0.03	0.056	**−0.93**	**0.87**	**−0.78**	**0.59**	**0.66**
L	—	**−0.80**	−0.23	0.04	−0.06	0.28	−0.36	0.39	−0.42	−0.48
J	—	—	−0.33	0.02	−0.04	0.38	−0.47	0.49	−0.46	−0.50
**PPS**	—	—	—	0.02	0.00	**−0.93**	**0.72**	**−0.67**	0.49	**0.59**
DP	—	—	—	—	**−0.99**	0.01	−0.07	0.07	−0.11	−0.07
SP	—	—	—	—	—	−0.04	0.11	−0.09	0.13	0.09
**ATP**	—	—	—	—	—	—	**−0.78**	**0.77**	**−0.59**	**−0.68**
**PL**	—	—	—	—	—	—	—	**−0.86**	**0.72**	**0.69**
NIQE	—	—	—	—	—	—	—	—	−0.73	−0.76
R	—	—	—	—	—	—	—	—	—	0.77

**Table 5 sensors-25-01009-t005:** This table presents the performance of the different methods used in the study expressed as a mean absolute error percentage. In bold we present the best results achieved in our experiments.

Algorithm	NIQE	FPS	R
Linear regression	2.68±2.051	9.93±11.448	10.83±12.875
Decision Trees	2.81±2.609	10.18±16.532	9.96±20.560
Random Forest	2.08±1.940	8.27±10.545	7.56±11.133
AdaBoost	2.68±2.051	10.24±12.368	11.09±13.013
XGBoost	3.04±2.687	10.68±15.082	10.88±22.823
SVM	2.39±2.183	8.68±11.308	8.48±12.547
QoENet1D	2.60±0.201	10.99±2.998	13.84±3.256
QoEAE1D	2.78±0.262	11.04±2.770	13.10±2.807

## Data Availability

The data and the code are freely available on https://github.com/mchlsdrv/qoe (accessed on 4 February 2025).

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
