# Peer review of "Estimating QoE from Encrypted Video Conferencing Traffic"

_sensors, 2025, doi:10.3390/s25041009_

Round 1

Reviewer 1 Report

Comments and Suggestions for Authors

Comments provided in attached file.

Author Response

Thank you very much for the insightful comments. We have tried to do our best to elaborate on all of the unclear points.

The attached file contains our answers.

Best regards,

Michael

Reviewer 2 Report

Comments and Suggestions for Authors

This paper addresses these challenges by analyzing a large dataset of Zoom sessions and training five classical machine learning (ML) models and two custom Deep Neural Network (DNN) to predict three QoE indicators. However, there are still some issues that need to be carefully addressed and revised.

1. The introduction needs to strengthen the analysis of the importance of traffic encryption to Internet security and its challenges to analysis and application. And how is data encryption reflected in this article? Please explain.

2. The paper mentions the analysis of a large number of Zoom conference datasets. Please provide a detailed explanation of the sources, collection methods, and preprocessing steps of the datasets. To enhance the reproducibility of the research, it is recommended to elaborate on the process of constructing the dataset.

3. The paper selected five classic machine learning models and two custom deep neural networks for prediction. However, please provide specific criteria for selecting these models, such as their performance on similar problems, computational efficiency, etc.

4. The experimental results section provides the mean square error rate of the model, and it is recommended to conduct in-depth analysis of the performance differences between different models, the performance of the model on different datasets, and the stability of the model's prediction results.

5. The literature in this article is too old, and there are even several articles published in 1901 and 1997. It is recommended to write more about the research status in the introduction section. At the same time, it is recommended to refer to the following relevant literature for comparative discussion.

[1] D. D. Hema and K. A. Kumar, Novel optimised deep learning approach for an efficient traffic state prediction based on CAE-ICCDCS-GRU model, International Journal of Bio-Inspired Computation, vol. 23, no. 2, pp. 8098, Jan. 2024.

[2] Lei Kou, Jinbo Wu, Fangfang Zhang, Peng Ji, Wende Ke, Junhe Wan, Hailin Liu, Yang Li and Quande Yuan, Image encryption for offshore wind power based on 2D-LCLM and Zhou Yi eight trigrams, International Journal of Bio-Inspired Computation, vol. 22, no. 1, pp. 5364, Jan. 2023.

[3] D. Yang and Y. He, Research on micro defect recognition based on deep learning, International Journal of Bio-Inspired Computation, vol. 23, no. 4, pp. 257265, Jan. 2024.

[4] S. S. George and R. S. Pramila, Improved whale social optimisation algorithm and deep fuzzy clustering for optimal and QoS-aware load balancing in cloud computing, International Journal of Bio-Inspired Computation, vol. 22, no. 1, pp. 4052, Jan. 2023.

Author Response

Thank you very much for the insightful comments. We have made every effort to clarify all the unclear points.

The attached file contains our answers.

Best regards,

Michael

Round 2

Reviewer 1 Report

Comments and Suggestions for Authors

Comments and Suggestions for Authors

The authors significantly improved the comprehensives and clarity of the article. However, the research is somewhat limited to cases with poor internet coverage in specific locations. Nonetheless, the presented idea is interesting and highly relevant to addressing current scientific problems related to end-to-end encrypted internet data traffic. In my opinion, the obtained research results can be scalable to other real-world applications. 

Some minor corrections required:

1.     Formatting starting from 13 page must be improved. To much empty space after table 4.

2.     Figures 5, 7, 13 are too big, takes too much place in respect to given information.

3.     Figure 7 – to big text size.

4.     Text size of figures 9 and 12 is too small. The text size must be the same in all graphs.

Author Response

Thank you for the review. Please find attached the reply. 

Reviewer 2 Report

Comments and Suggestions for Authors

accept

Author Response

Thank you for your previous comments.